# Changes in the Functional Role of the Tejo Estuary (Portugal, Europe) According to Fish Ecological Guilds

**Susana França** [1,2]

[1] MARE—Marine and Environmental Sciences Centre/ARNET—Aquatic Research Network, Faculdade de Ciências, Universidade de Lisboa, 1749-016 Lisboa, Portugal; sofranca@fc.ul.pt
[2] Departamento de Biologia Animal, Faculdade de Ciências, Universidade de Lisboa, 1749-016 Lisboa, Portugal

**Abstract:** Estuaries are extremely productive ecosystems, providing habitats for numerous aquatic species and crucial ecological services. The Tejo estuary, one of the largest European estuaries, has been thoroughly studied, and its important functional role as a nursery for several commercially important fish species is already established. In the present work, a trait-based approach was applied to functionally describe the fish community structure of the Tejo estuary and to enlighten potential changes in the ecosystem functioning at this level, following environmental changes expected to occur. To predict the distribution of species from the two most representative ecological guilds of the Tejo estuary, estuarine residents and marine migrants, species distribution models were built using an ensemble technique (combining forecasts of single models). The predictions obtained were more accurate for the marine migrants and the species distribution was strongly related with salinity, whereas estuarine species, were also influenced by depth, habitat type and river flow. The potential distributions of these ecological guilds showed that marine migrants will tend to use upstream areas in the estuary, where salinity is lower. Nonetheless, salinity is expected to increase as extreme weather events such as droughts tend to occur more frequently, decreasing favorable habitat availability for these species, and thus threatening the crucial role this ecosystem plays for these species.

**Keywords:** estuaries; fish ecological guilds; species distribution models; ensemble modeling; global changes

**Key Contribution:** Species distribution models results indicate that the crucial role the Tejo estuary plays as a nursery for several commercially important fish species may be threatened in the face of future global changes.

## 1. Introduction

Estuarine ecosystems are among the most biologically productive ecosystems on Earth [1], providing critical ecological services such as coastal protection, nutrient cycling and maintenance of biodiversity and nursery grounds for many fish species of ecological and economic importance [2,3]. Nonetheless, they are also ranked amongst the most impacted ecosystems, with continuous infrastructure development and the longstanding population growth increasing pressure in a multifaceted way [4]. Simultaneously, and at a global scale, conservation measures at this level continue to be inefficient, failing to target sensitive fish species distributions, which remain susceptible to high pressures [5].

The highly variable and widely fluctuating estuarine environmental conditions are challenging for most marine and freshwater fish species, and only those presenting a high tolerance to these factors have been able to benefit from these productive natural ecosystems at their highest potential [6]. These local environmental variations are mainly responsible for the low diversity of estuarine fish assemblages, which are generally composed of marine species that migrate into estuarine systems for reproduction, rearing or maturation. They

may also take advantage of the estuarine resources, use estuaries as migration corridors between rivers and oceans, or spend part of their life cycle in those areas [7,8].

As estuaries have their own specific features and dynamics, they present differences in terms of either biotic or abiotic characteristics, with fish assemblages also expected to differ. Nonetheless, if fish species in estuaries would present consistently similar responses to the environment, assemblages would be expected to reflect this similarity [9], particularly at a functional level. Therefore, trait-based approaches are often used to better understand community assembly rules and ecosystem functioning [10] and may be of great utility when trying to infer the response of estuarine fish communities to changes in environmental conditions at a functional level [11]. Thus, Elliott et al. [12] developed the concept of estuarine functional guilds, which was later developed by Potter et al. [13], and successfully used to understand spatial and temporal variation patterns of fish assemblages [14], to identify changes in the food webs structure and energy flow of estuarine systems [15], to assess migratory routines and physiological adaptations of species using estuaries throughout their life cycle or part of it [12], and to understand the effects of climate changes on the structure and composition of fish fauna [8].

Due to their location, estuaries and coastal regions are expected to be severely impacted by climate change and by the occurrence of extreme weather events [16,17], making it crucial to improve knowledge on how estuarine fish communities will respond to effectively manage issues in these areas. Several works have already been conducted for freshwater ecosystems and the results have shown that a global decline in fish species due to climate change represents a major conservation challenge in these domains [18]. Similar results in comparable assessments were obtained for coastal fish assemblages, with an expected decrease in species richness, while sea surface temperature increased [19]. This kind of projection is scarcer for estuarine-dependent fish at a functional level. For instance, Martinho et al. [20] showed that the depletion of freshwater species, the increase in marine adventitious species, and the significant reduction in abundance of truly estuarine species at the Mondego estuary (Portugal) reflects the effects of severe droughts in the region, which are predicted to occur more frequently in the future. Salinity regimes have also been shown to be a major driver in determining which fish ecological groups dominate estuaries, with marine migrants, i.e., fish species using estuaries as nurseries, presenting a climate-dependent life cycle, particularly during their early life stages [21,22].

Numerical models have been shown to be robust and adequate tools that allow us to thoroughly analyze estuarine dynamics and to consistently forecast the impacts of anthropogenic activities, extreme weather events, and climate change conditions [23]. Estuaries and adjacent areas along the Portuguese coast have been studied in detail in recent decades, and their crucial nursery role and habitat use by juveniles of several commercially important species have been frequently highlighted [24,25]. Existing time-series on physical parameters have already supported some preliminary forecast exercises on expected changes in some Portuguese estuaries [26], but available data for biological elements are much less comprehensive. Nonetheless, numerical models may help to surpass the lack of field observations, providing valuable information to promote these ecosystem's conservation and health [27]. At a global level, these tools have been implemented to represent the effect of climate change on estuarine morphology and hydrodynamics, e.g., [28–30], whereas at an ecological level, they have been used to develop species distribution models for commercially important fish species using estuaries as nursery grounds [31–34] and to assess global patterns and predictions of species richness variation [35,36].

The Tejo estuary is one of the largest European estuaries and has been thoroughly studied, and its important ecological role, namely acting as a nursery for several commercially important fish species, is already established [24,37,38]. It is a highly explored system, and has long been affected by strong and varied human activities such as industrial development, urbanization and port and fishing activities [39,40]. Due to its geographic location (near the southern and northern limits of distribution of many marine and estuarine fish species), the Tejo estuary is a good model to investigate and understand the effects of

environmental changes in its fish communities, namely at the functional level. Therefore, the present study uses a wider database from the Tejo estuary, gathering data from research projects conducted over the last two decades, and ensemble modeling techniques to predict how estuarine fish ecological guilds (estuarine species and marine migrants) distribution will change according with the influence of several environmental factors. Ultimately, the models' predictions will allow us to explore the hypothesis as to how the main estuarine functions will change according to the most frequently anticipated climate changes.

## 2. Materials and Methods

### 2.1. Study Area

The Tejo estuary is one of the largest estuaries on the Atlantic coast of Europe, with a length of 50 km and an area of 325 km$^2$ (Figure 1), of which about 40% is intertidal areas, predominantly mudflats fringed by extensive areas of salt marshes. It is a partially mixed estuary with a tidal range of about 4 m. It is a mesotidal system with semidiurnal tides and with a complex geomorphology: its main channel comprises a narrow inlet with 4 km width and depths from 25 to 30 m, linking the Atlantic Ocean to a broad inner basin of 10 km width at its maximum, and depths ranging from 0 to 12 m in the mixing region. At the upper reaches, where the large intertidal mudflats and salt marsh areas are located, the estuary's depth is lower than 1 m [41–43], making the overall estuarine mean depth lower than 10 m. The river flow varies markedly either seasonally and interannually, presenting a mean value of 400 m$^3$ s$^{-1}$. Salinity varies from 0 at the estuary upstream limit (50 km from the estuary's mouth) to nearly 37 at the mouth. The estuary bottom is composed of a heterogeneous assortment of substrates: its main sediment is muddy sand in the upper and middle estuary and sand in the lower estuary. The Tejo estuary includes several protected areas, one of the most important being the Tejo Estuary Nature Reserve, with an area of 145 km$^2$. Most of this area is composed of intertidal mudflats that emerge at low tide, allowing wintering birds that use the area as a stopover site during their winter migrations to feed [44].

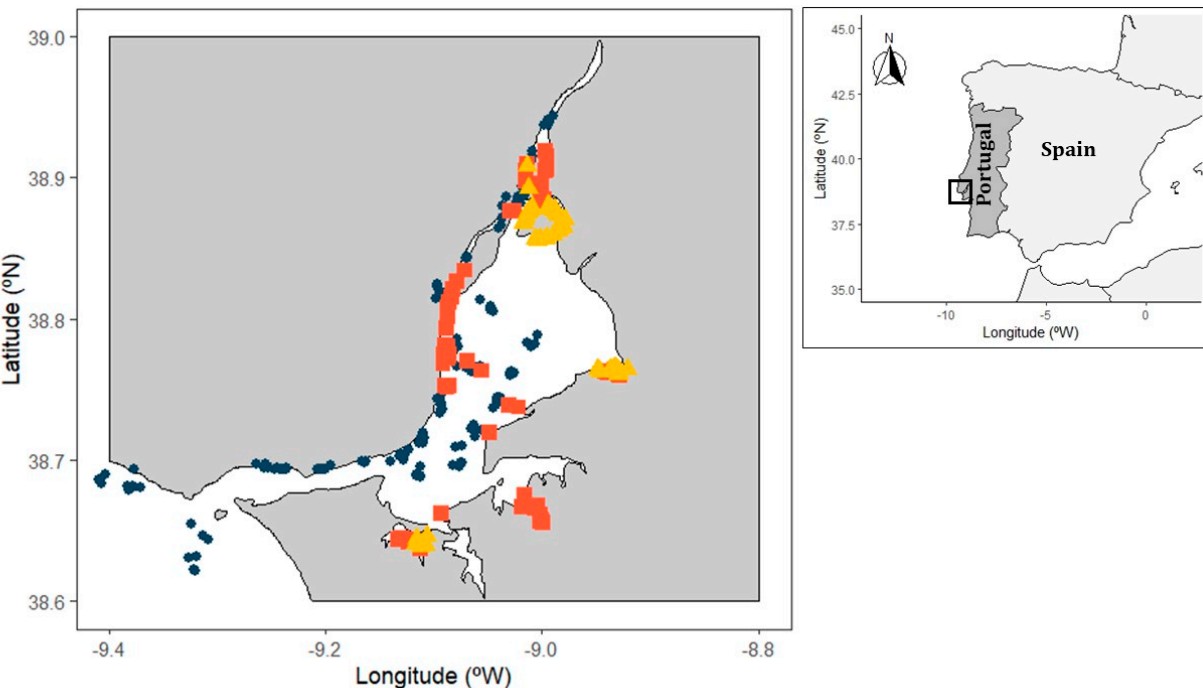

**Figure 1.** Location of the Tejo estuary. Symbols represent sampling points from the three projects analyzed. Blue circles: sampling from project 1 (2001 and 2002); orange squares: sampling from project 2 (2006); yellow triangles: sampling from project 3 (2009).

### 2.2. Sampling

In order to obtain a wider temporal database for fish species using the Tejo estuary, data were taken from the historical data section of the CoastNet Research Infrastructure geoportal (geoportal.coastnet.pt, accessed on 23 June 2023), which contains a compilation of data collected during 15 research projects for Portuguese estuaries. To avoid dealing with different sampling methodology constrains, data were obtained from 3 projects conducted in the Tejo estuary using the same sampling methods and materials: (1) In the first project (ERIC), sampling was conducted between May 2001 and March 2002, every two months. Sampling sites were determined according to salinity and located throughout four areas in the Tagus estuary, with 10 hauls conducted at each area, covering the whole estuarine gradient. Total number of samples: 240 hauls. (2) In the second project (NURSERIES), sampling was carried out in May and July 2006, with 10 hauls performed per site each month, and 10 sites along the main nursery grounds across the salinity gradient. Total number of samples: 200 hauls; (3) in the third project (FISHEST), sampling was habitat-oriented, focusing on salt marsh, intertidal mudflats and subtidal channels (see classification, [45]). Fish assemblages were sampled in January, April, July, and October 2009, to cover the seasonal gradient, in two sites per estuary. At each of these sites, three hauls were carried out in each of the three habitats. To avoid overlap sampling between different habitats, hauls were restricted to a specific habitat type. Total number of samples: 52 hauls.

For all the projects, sampling surveys were conducted using a beam trawl with a tickler chain and 5 mm mesh size in the cod end. Hauls were performed at a constant speed ($0.8$ m s$^{-1}$) and lasted between 8 and 10 min, whenever there was sufficient habitat area available. Salinity, conductivity (ms cm$^{-1}$), water temperature (°C) and dissolved oxygen (% saturation) were measured at the beginning of each haul, using a multi-parameter probe (WTW ProfiLine Oxi 197, Weilheim, Germany) and depth (m) was also registered. Sampled area (m$^2$) was calculated based on the trawl opening width (2 m) and the distance traveled (m) (obtained using the coordinates registered at the beginning and at the end of each haul, with a global positioning system (GPS)). All fish caught were identified, counted, and weighed.

### 2.3. Data Analysis

Species richness, which implies counting species without considering the number of individuals, meaning that equal weight is given to all of them, was calculated for each sampling year. Afterwards, each fish species was assigned to a functional guild according to Franco et al. [46]. The ecological guilds considered truly estuarine species (ES: may complete their life cycle within the estuary, meaning that may breed in these ecosystems; highly euryhaline species, may adapt to the estuarine conditions and move throughout the full estuarine area), marine migrants (MM: spawn at sea and regularly, a large number of individuals enter estuaries; highly euryhaline species, able to move throughout the full length of the estuary, including marine species, with juveniles seasonally using estuaries as nursery grounds, and migrating seawards when their sub-adult stage is attained), marine stragglers (MS: species that spawn at sea; usually associated with coastal marine habitats, with few individuals entering estuaries accidentally; predominantly stenohaline species, occur most frequently in the estuary lower reaches), freshwater species (F: spawn in freshwaters, entering regularly into estuaries, in moderate numbers, able to move down estuaries at varied distances), anadromous species (A: live in the sea but regularly use estuaries as migration corridors to freshwaters, where reproduction occurs); and catadromous species (C: live in freshwater but regularly use estuaries as migration corridors to the sea, where reproduction occurs). According to the species assigned to each ecological guild, mean density, which reflects the mean number of individuals, and biomass were determined and expressed in individuals 1000 m$^{-2}$ and g 1000 m$^{-2}$, respectively, for each guild and for each sampling year.

Ensemble Distribution Models

Distribution models were created for the estuarine species (ES) and marine migrants (MM) ecological guilds. Species assigned to the marine stragglers (MS) guild enter the estuary accidentally and often correspond to a single individual caught near the estuary's mouth. These species are not able to cope with the estuary's dynamics and environmental variability and leave soon after entering. For these reasons, and because their spatial distribution was confined to a restricted area, downstream, distribution models were not created for this guild. The remaining ecological guilds were not representative enough to build and run robust distribution models generating relatively accurate predictions. Models were built to relate and predict the species ecological guilds' occurrence with the previously described environmental variables measured for the sampling, position, and habitat type of each tow, as well as the mean river flow and precipitation obtained for each sampling month and year (obtained in https://snirh.apambiente.pt/, accessed on 23 June 2023). The environmental variables used as predictors in the models were chosen based on their known relationship with estuarine fish species, and their ability to influence these species distribution in estuaries, which has been previously assessed and thoroughly discussed [21,24,32,33,47].

Problems of collinearity (or multicollinearity) may arise when a strong correlation between explanatory variables in a regression model occurs, and their individual influence on the response variable becomes difficult to distinguish [48]. Variance inflation factors (VIF), which measure the degree of inflation of the unexplained variance due to the inter-correlation among independent variables, were used to deal with collinearity issues at variable selection [49]. Variables with a VIF higher than four were excluded.

To model the potential distribution of the most represented ecological guilds in the Tejo estuary, four different algorithms from the BIOMOD2 package [50] were used in R software (version 4.2.3) [51]. The algorithms used were GLM (Generalized Linear Models), GAM (Generalized Additive Models), RF (Random Forest for Classification and Regression) and CTA (Classification Tree Analysis) (Appendix B). The models were evaluated by splitting data into training and testing datasets. To reduce model overfitting, the obtained datasets should ideally be spatially independent [52,53]. The "block" method was used to partition data based on the latitude and longitude of the sampling points, dividing the occurrence localities as equally as possible. K models were run iteratively, using k−1 parts for training and the remaining one for testing.

An ensemble model was obtained for these ecological guilds in the Tejo estuary by averaging the four final predictions. For each of the ensemble models, three measures of model performance were assessed: (i) AUC: measures the area under a receiver operating characteristic (ROC) curve, which plots sensitivity against (1-specificity) over a number of classification thresholds. AUC is threshold independent and evaluates the models' ability to correctly predict a higher probability of occurrence where species are present than where they are absent. Values around 0.7 indicate poor model performance, as they suggest similar rates of correct and erroneous predictions; values from 0.7 to 0.9 indicate (moderately) useful models; and values exceeding 0.9 signify excellent accuracy [54]. (ii) Sensitivity quantifies the proportion of correctly predicted observations of species presence. The sensitivity is low, therefore, when omission errors (erroneous predictions of absence) are common, in addition to (iii) specificity, which conversely, identifies the proportion of correctly predicted observations of species absence. Thus, specificity is low when commission errors (erroneous predictions of presence) are frequent [55]. To calculate the two measures, probabilistic estimates of species occurrence must be classified into presence–absence predictions. To transform the obtained probabilities into presences and absences, a threshold needs to be used and its value may be calculated using a wide range of methods, whose selection can have dramatic effects on model accuracy. Although it is the most used threshold, the default value of 0.5 does not necessarily give the highest prediction accuracy, especially for datasets with very high or very low observed prevalence [56]. In the present study, the threshold at which the predicted prevalence is equal to the observed prevalence

was used, with previous studies showing that higher values of accuracy measures were registered with its use [34].

## 3. Results

### 3.1. Diversity Metrics

A total of 66 fish species were collected during the sampling surveys (three projects conducted in 2001, 2002, 2006 and 2009) (Appendix A). Each species was assigned to an ecological guild, whose proportions varied considerably. Figure 2a shows that marine stragglers (MS) was the dominant guild, comprising 57.6% of the species, when analyzing the whole fish community for the four sampling years, followed by marine migrants (MM) with 19.7%, estuarine species (ES) with 16.7%, catadromous species (C) with 4.5% and freshwater species (F) with 1.5%.

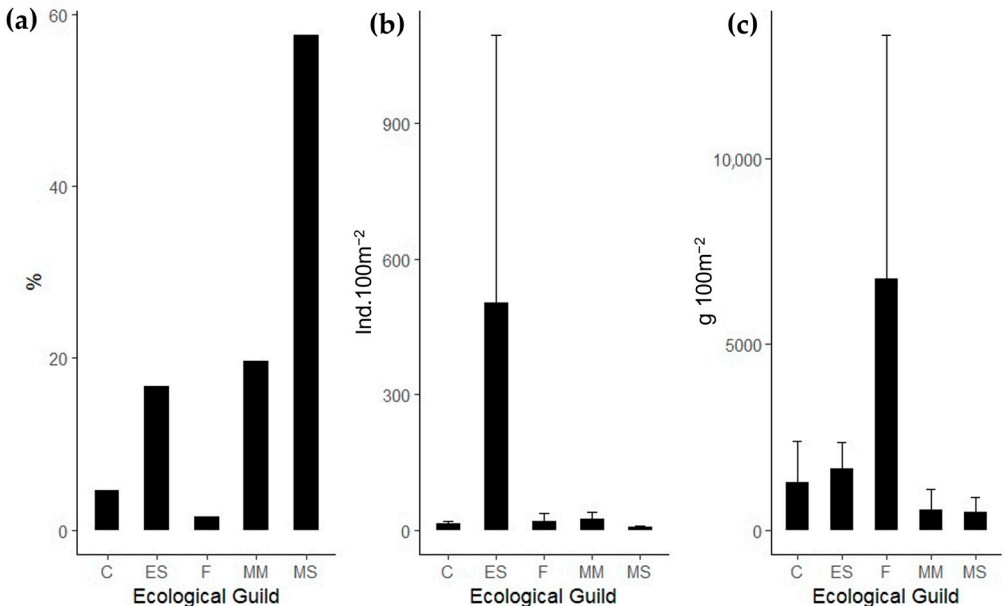

**Figure 2.** (**a**) Percentage of species from the ecological guilds considered, in the Tejo estuary; (**b**) Mean density (and standard deviation) for each ecological guild; (**c**) Mean biomass (and standard deviation) for ecological guild. Ecological guilds considered—C: catadromous species; ES: Estuarine species; F: freshwater species; MM: marine migrants; MS: marine stragglers.

The fish assemblage from the Tejo estuary was also characterized according to the mean densities and biomasses of the species ecological guilds, as the species richness previously calculated does not take them into account. Although most of the species were marine stragglers, the highest mean density was found for estuarine species with 502.2 ind.1000 m$^{-2}$, followed by marine migrants with 23.9 ind.1000 m$^{-2}$ (Figure 2b). Although in lower numbers, freshwater and catadromous species presented higher densities (19.6 ind.1000 m$^{-2}$ and 13.9 ind.1000 m$^{-2}$, respectively) when compared with marine stragglers (5.5 ind.1000 m$^{-2}$), which presented a considerably higher number of species (Figure 2b).

Although present in the fish assemblage of the Tejo estuary with the lowest number of species, the freshwater ecological guild presented the highest mean biomass, 6767.4 g 1000 m$^{-2}$, meaning that even in low numbers, individuals from these species are the heaviest (Figure 2c).

### 3.2. Ensemble Models

3.2.1. Model Performance and Variables Importance

Table 1 presents the values of the accuracy measures AUC, sensitivity and specificity obtained for the ensemble distribution models built for estuarine and marine migrant species. The accuracy values obtained for the model of the marine migrants were slightly higher, indicating an overall better predictive performance than the resident species model. The latter presented a moderate accuracy in accordance with its AUC (0.79). The capacity for correctly predicting observations of species presence was relatively high, with sensitivity presenting a value of 0.79, while specificity was lower (0.67), indicating lower accuracy with regard to predicting species' absences (Table 1). For the marine migrant species model, the AUC was 0.9, revealing moderate accuracy (close to excellent). The values obtained for sensitivity (0.84) and specificity (0.7) were also higher than the ones obtained for estuarine species, which indicates that the capacity to correctly predict either presences or absences is better when the model is applied to marine migrant species (Table 1).

**Table 1.** Accuracy measures obtained for the ensemble distribution models of estuarine and marine migrants' species from the Tejo estuary.

|  | AUC | Sensitivity | Specificity |
|---|---|---|---|
| Estuarine Species | 0.79 | 0.79 | 0.67 |
| Marine Migrants | 0.90 | 0.84 | 0.70 |

All the variables were used as predictors when building the models, as no collinearity was registered when calculating the VIF (all predictors with VIF > 4).

Amongst all the environmental variables considered in the final model, depth, habitat type, river flow and salinity were considered the most important predictors when predicting the ES distribution (Figure 3), achieving relative values higher than the mean importance value obtained for all the variables. Overall, presences of estuarine resident fish species are expected to occur at lower depths within the estuary, though there are weaker differences between presences and absences at lower values of river flow and slightly higher salinities (Figure 4).

When analyzing variables' importance within the ensemble distribution model of marine migrants, salinity was clearly the most important predictor, presenting the only relative value higher than the mean importance values obtained for all the predictors (Figure 5). According to Figure 6, marine migrant species can be found at wider spans of salinity, which include lower values than the ones obtained for these species' presence.

3.2.2. Current and Potential Future Distributions of Ecological Guilds

The potential distributions of resident and marine migrants' species under current and future scenarios, for the Tejo estuary, are presented in Figures 7 and 8. When comparing the predictions made by the distribution ensemble model for the resident species, with the presences observed with the projects' sampling, no large differences are found. Overall, the predictions map for resident species distribution shows that they slightly decrease downstream and in the innermost bay, which may relate to the depth (the deeper sites are found close to the estuary's mouth), and habitat type (habitats such intertidal mudflats and saltmarshes are common in the estuary's enclosed bays), two of the most important variables related to the resident species' presence in the estuary.

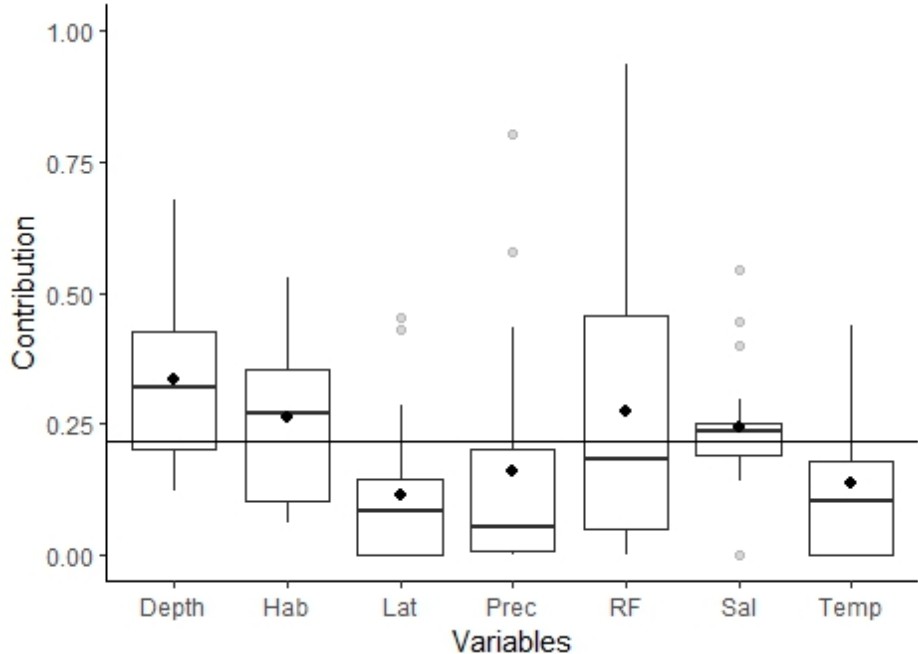

**Figure 3.** Contribution of each variable to the final ensemble model predicting the distribution of truly estuarine species (ES) in the Tejo estuary (for each boxplot—black line: median value; black circle: mean value; box value: 1st and 3rd quartiles; end of outer lines: minimum and maximum; circles outside the boxplot: outliers). The continuous line represents the global mean value of the contribution of all the variables. The variables considered are depth; habitat type; latitude; precipitation; river flow; salinity; and temperature.

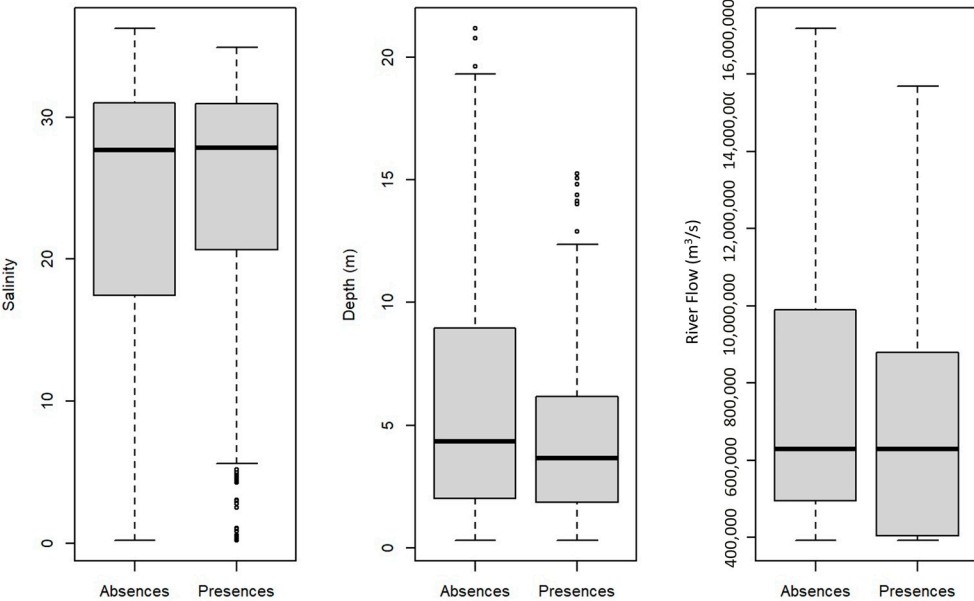

**Figure 4.** Influence of the most important variables in the occurrence of estuarine species in the Tejo estuary (for each boxplot-middle line: median value; box value: 1st and 3rd quartiles; whisker values: minimum and maximum; circles outside the boxplot: outliers).

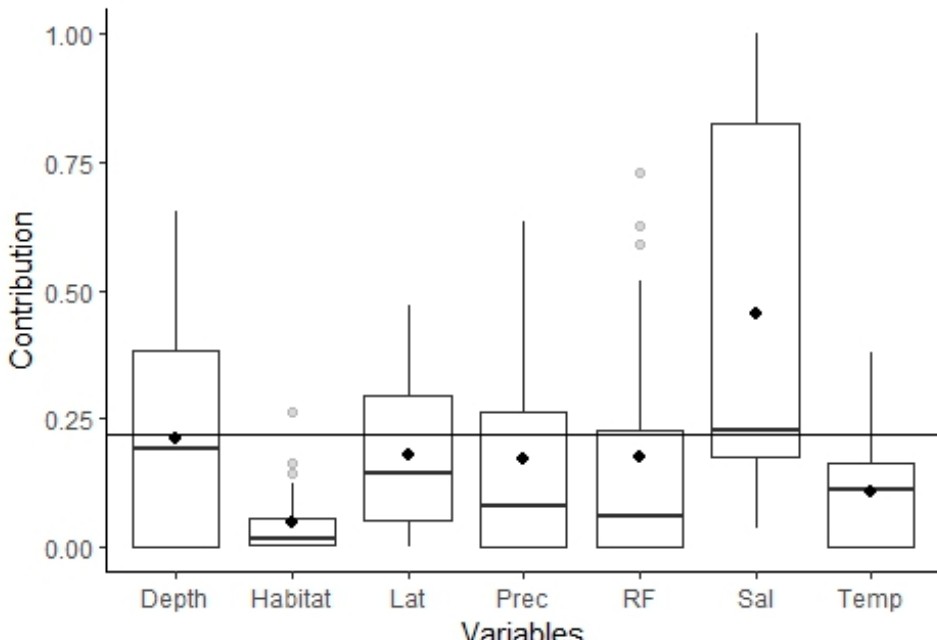

**Figure 5.** Contribution of each variable for the final ensemble model predicting the distribution of marine migrants species (MM) in the Tejo estuary (for each boxplot–black line: median value; black circle: mean value; box value: 1st and 3rd quartiles; end of outer lines: minimum and maximum; circles outside the boxplot: outliers). The continuous line represents the global mean value of the contribution of all the variables. The variables considered are depth; habitat type; latitude; precipitation; river flow; salinity; and temperature.

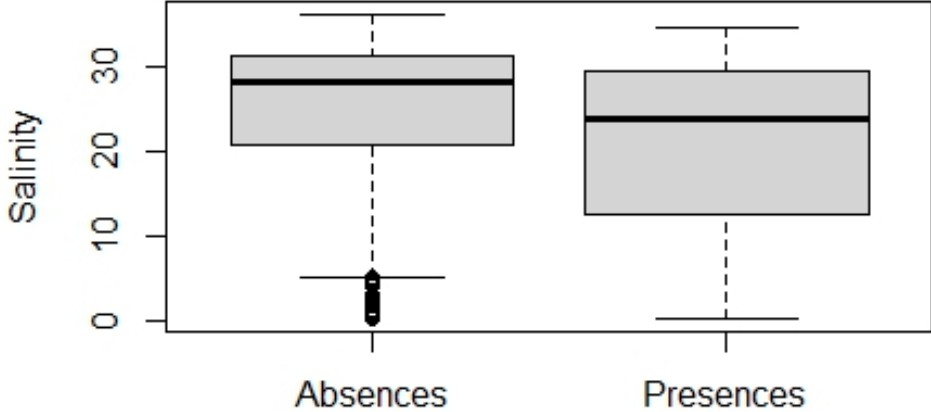

**Figure 6.** Influence of salinity in the occurrence of marine migrants' species in the Tejo estuary (for each boxplot—middle line: median value; box value: 1st and 3rd quartiles; whisker values: minimum and maximum; circles outside the boxplot: outliers).

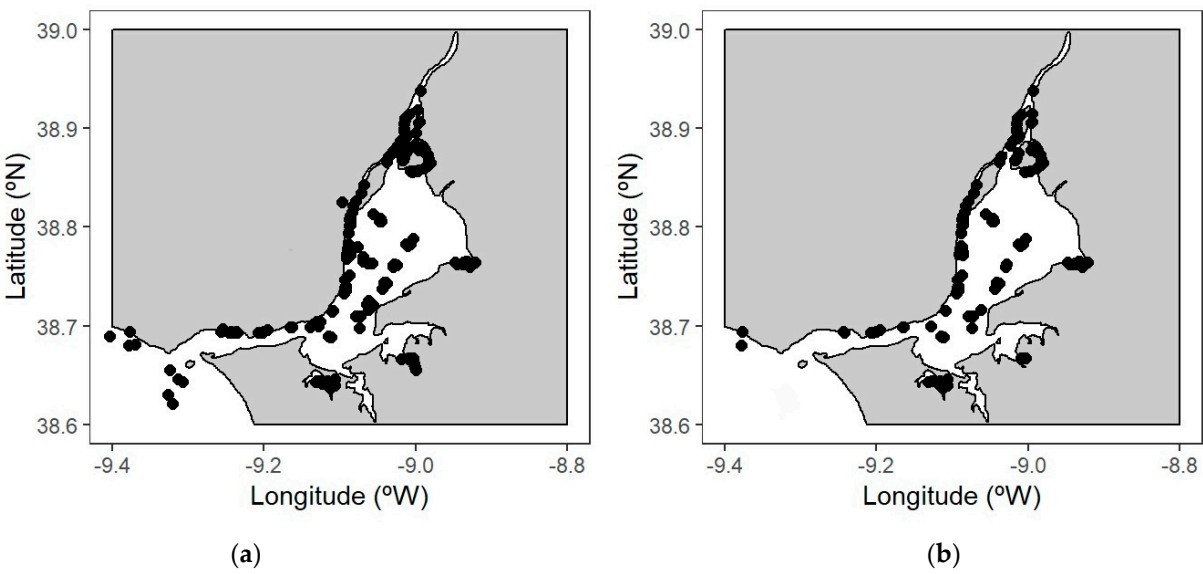

**Figure 7.** (**a**) Observed presences for estuarine species, obtained from sampling in the Tejo estuary; (**b**) predictions of estuarine species presence obtained from the ensemble model built for this ecological guild in the Tejo estuary.

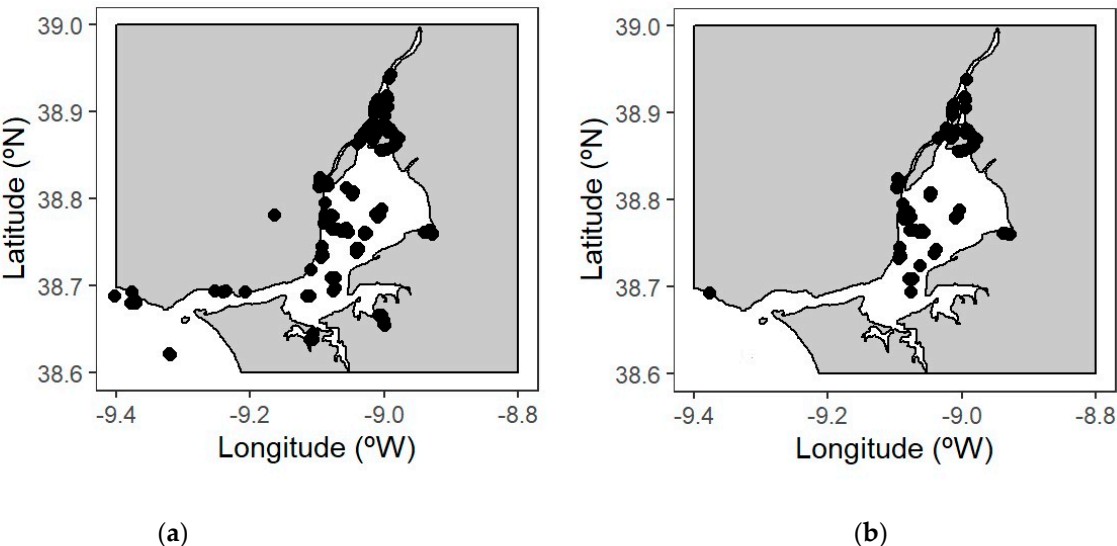

**Figure 8.** (**a**) Observed presences for marine migrants' species, obtained from sampling; (**b**) predictions of marine migrants' species presence obtained from the ensemble model built for this ecological guild, in the Tejo estuary.

Regarding the marine migrant species, marked differences between the observed presences and predictions are found (Figure 8): a significant decrease in these species' presence is found in the most downstream areas, where the effect of higher salinity values can be found, while no such differences are found upstream, where the influence of the freshwater is stronger.

## 4. Discussion

### 4.1. Fish Assemblages from the Tejo Estuary: Ecological Guild Composition and Relation with Important Environmental Variables

Overall, fish ecological guilds composition for the Tejo estuary was similar to that of the fish fauna typically found in other European estuaries, with marine stragglers com-

prising more than 50% of the species caught, followed by marine migrants and estuarine species [20,57,58]. In fact, marine stragglers are invariably represented in estuaries throughout the world, whereas species that truly depend on estuaries are far less common [13]. Although it might seem that much of the Tejo estuarine fish assemblage originates from marine areas, marine stragglers enter the estuary in low numbers and their presence can be attributed to an uninterrupted connectivity between the system and the coastal area, through a large estuarine mouth, which allows them to explore the estuary opportunistically [36]. Estuarine species clearly dominated in terms of abundance, which corroborates previous findings in which this group comprised about 50% of the estuarine fish community [8,20,59]. Furthermore, the relatively high abundance of estuarine dependent species, such as marine migrants, confirm the dependency between the estuary and its adjacent coast [14]. Species within these dominant guilds use different habitats according to their life stages [37], presenting physiological and morphological adaptations that allow them to support the challenging condition of these areas, such as high plasticity in their diet [8,60] and wide salinity tolerance [12]. In contrast, marine stragglers enter estuaries in low numbers, often accidentally, and their presence is typically brief, as they tend to lack these adaptations [17,61]. Freshwater species dominated the biomass of the assemblage, although their abundance was extremely low, representing a low number of larger individuals. Although the structure of estuarine fish assemblages is fairly comparable in temperate estuaries, regional variances may occur in the composition and abundance of their ecological guilds, which may be attributed to the specific characteristics of each estuarine system, or even seasonally, depending on the environmental conditions, namely the balance between freshwater flow and marine waters [45].

The use of ensemble modeling techniques to forecast species distribution has often been used to reduce model-based uncertainty [33,34,62–64]. Such an approach allows us to reduce potential errors due to overfitting and avoids the need to select the single best model among several similar ones [65]. The predictive performance of the ensemble models built for the main ecological guilds of fish species from the Tejo estuary was in line with the values previously obtained for fish species distribution for the Portuguese estuaries [33,34]. Although none of the models achieved values of excellent accuracy, predictions obtained for the marine migrants' ecological guild were more accurate than the ones for estuarine species. Similar results were obtained by França and Cabral [34], highlighting that when a fair representation of the important seasons and habitat types for these species is considered, the models' accuracy may be improved, along with their reliability.

The influence of natural and anthropogenic stressors on the structure of estuarine fish communities, as characterized by guilds, has yet to be fully quantified.

Depth, river flow, habitat type and salinity were the most important environmental descriptors determining the share of variance in the distribution of truly estuarine species. Previous studies have shown that depth does play a role in structuring estuarine fish assemblages, either directly, as often high densities of fish species may be found in shallow waters [66], or indirectly, due to ecological interactions with other species (e.g., competition and predation) and with other environmental variables such as temperature [67]. In the present study, truly estuarine species seem to be present at lower depths. Estuarine residents in the Tejo estuary are largely composed of the common goby, *Pomatoschistus microps* [33,37], a relatively small-bodied species, which due to their dimensions may largely benefit from shallow sheltered areas within these systems [68].

In the field of estuarine ecology, river flow has long been regarded as one of the strongest factors influencing fish species' presence and distribution [21,69,70]. The presence of estuarine species in the Tejo estuary seems to be related to lower values of river flow, and while these species complete their life cycle within these systems, it has been previously observed that strong variations in river flow may adversely impact estuarine larval fish assemblages [70]. In fact, the opposite results were found in the Mondego estuary (Portugal), which presented abundance peaks of resident species in years with higher precipitation and river runoff levels [71], and consequently, lower salinity values, which

has been shown to be an advantage for these species' egg development, as their survival is lower at higher salinities [72]. Differences found for this guild, between the Mondego and the Tejo estuaries, may be attributed to their different specific geomorphological and hydrological characteristics. The Tejo estuary has consistently higher values of river flow when compared to Mondego, and its configuration with one large main channel may contribute by flushing out larvae with weak swimming ability or blocking their assess to suitable estuarine habitats, which, in the end, may affect these species' life cycle closure [73].

Previous studies have already demonstrated that habitat type is an important feature influencing several species distribution patterns, namely truly estuarine ones, within the Tejo estuary [21,33,37]. Accordingly, each estuarine habitat type may have its own typical fish assemblage, which has been already described for Portuguese estuaries: the saltmarsh fish assemblage was found to be mainly composed of truly estuarine species, such as the common goby *P. microps*, and the same species accounted for a large proportion of the total number of fishes collected in the intertidal mudflat of the Tejo estuary [37]. The same study revealed that estuarine species were present in all the habitats considered but their proportions differed in each of them, dominating the intertidal and subtidal soft substratum, the saltmarsh and the freshwater habitats [37]. Shallow vegetated areas, such as saltmarshes, are also considered to be important estuarine habitats, providing feeding grounds and development and protection refuges for fishes, namely estuarine species. In fact, a recent analysis of a two-decade data series revealed a declining trend in the abundance of an estuarine resident fish, which was associated with the historic decrease in submerged vegetated meadows in Patos Lagoon, Brasil [74].

For marine migrants, the effect of salinity seems to be considerably stronger when determining these species distribution, compared to other environmental variables. In fact, this variable has long been regarded, globally, as an extremely important factor in structuring estuarine fish communities, as species occurrence results from their differential tolerance to it [75,76], and for marine migrant species, namely in Tejo [21,33]. Similar results have been found by da Silva et al. [77], with no seasonal effects on species composition of marine migrants, nor a correlation with environmental conditions. These authors associated these findings with the species life cycle, as they use estuaries as juveniles and remain there until they are ready for recruitment, not being affected by seasonal or environmental changes during that time. On the contrary, the solely estuarine species had significant changes associated with several variables, such as rainfall and habitat type, indicating that the coastal dynamics allow for the whole habitat mosaic to be available for these species, which may use them at different scales, spatial and seasonally. These spatial and temporal segregations allow for a reduction in species' competition for resources, while encouraging various species to perform their functions in different habitat types and seasons [63,78].

*4.2. The Tejo Estuary Functioning in Face of Global Changes*

The increase in global mean temperature throughout the last 100 years, the register of the ten warmest years in the last three decades and the years 2005 and 2010, which were highlighted as the warmest years of the century, are amongst several indicators that exhibit climate change unfolding [79]. Extreme weather events are already an established effect of climate change and events such as heavy precipitation, persistent droughts and frequent heat waves are expected to continue to increase in the next few years [80]. Global changes are thus a well-established reality for aquatic systems and consequences may translate in expected rapid changes in water temperature and salinity in estuarine systems, which will most probably alter fish community abundance, structure, composition and distribution [80].

Given the threat that climate change poses for the global functioning of estuaries, it is critical to understand how it will impact future distributions of the main fish ecological guilds. The models built in the present study have predicted that the distribution of the resident fish species of the Tejo estuary will present slight changes, with fewer fish appearing downstream, just outside the estuary's mouth and in one of its inner bays. These

small differences may be related with depth (deeper sites are found close to the estuary's mouth, and estuarine species tend to prefer shallow habitats), and habitat type (habitats such as intertidal mudflats and saltmarshes are common on the estuary's enclosed bays), two of the most important variables related to the resident species presence and distribution within the Tejo estuary.

For marine migrants, the model predicted a potential reduction in their suitable areas within the Tejo estuary, with these species occurring mostly upstream, in areas with a stronger influence of freshwater. Extreme weather events are now predicted to occur more frequently and at stronger intensities as a consequence of climate change [79,81], with longer and more frequent periods of drought expected to occur for the Portuguese coast [82]. In transitioning ecosystems such as estuaries, the salinity gradient may change according to the precipitation regimes. Therefore, drought years will decrease the freshwater input in the system, either through less frequent precipitation episodes or due to lower values of river flow. According with the present study, marine migrants will tend to be absent at higher salinity areas, which in drought years may increase throughout the Tejo estuary, jeopardizing the crucial ecological role it plays as a nursery area for juveniles of several fish species, namely some commercially important ones. In the Mondego estuary (Portugal), estuarine residents and marine migrants (e.g., European sea bass, European flounder and common sole) increase their abundance with higher rainfall/runoff due to the increased productivity, as both processes are strongly related, benefiting coastal fisheries landings from these species [83]. Furthermore, these expected reductions in the intensity and frequency of rainfall will decrease the river flow effect on the coastal areas, possibly affecting the recruitment of several marine migrant species, due to the reduction in the extent of river plumes in the estuary adjacent coastal areas [84].

## 5. Conclusions

This study shows that the functional structure and trait composition of fish in the Tejo estuary vary according to different environment variables, namely salinity, which may pose some challenges in face of the global change scenarios envisioned. Specifically, the importance of the Tejo estuary as a crucial nursery area may be threatened by extreme weather events such droughts, which will increase salinity intrusion in the estuary, decreasing favorable habitat availability, as marine migrants seem to prefer moving towards more upstream habitats, where the salinity is usually lower.

Although the present study tries to forecast species ecological guilds' future trends based on as many data as possible from the past, which in this case includes data from three research projects conducted throughout the past two decades, this may not be sufficient to detect slight yet important changes in estuarine fish community composition, and thus in the global ecosystem functioning. These changes are not always easy to record, due to the dynamic nature of these species, their interactions, and the environment itself. Therefore, long-term monitoring of the estuarine fish community, combined with climate data, would be crucial to detect these changes and trends in relation to various stressors. Results from the present study and the predicted climate trends should be strongly considered when undertaking management plans for estuaries and river catchment areas.

**Funding:** This research was funded by Fundação para a Ciência e a Tecnologia (FCT, I.P.) through the Strategic Project granted to MARE—Marine and Environmental Sciences Centre (UIDB/04292/2020 and UIDP/04292/2020) and the project granted to the Associate Laboratory ARNET (LA/P/0069/ 2020). Data for this publication were obtained via the CoastNet Research Infrastructure geoportal (http://geoportal.coastnet.pt, accessed on 23 June 2023). CoastNet was funded by FCT, I.P. and the European Regional Development Fund (FEDER), through LISBOA2020 and ALENTEJO2020 regional operational programmes, in the framework of the National Roadmap of Research Infrastructures of strategic relevance (PINFRA/22128/2016). This study had also the support of "DANUBIUS-IP—DANUBIUS Implementation Phase Project", HORIZON-INFRA-2021-DEV-02-02-101079778, financed by European Union's Horizon Europe research & innovation programme under

the Grant Agreement No. 101079778. Susana França was supported by FCT, I.P., via research contract CEECIND/02907/2017.

**Data Availability Statement:** Publicly available datasets were analyzed in this study. These data can be found at the "Historical data" section, from the CoastNet Research Infrastructure geoportal (http://geoportal.coastnet.pt, accessed on 23 June 2023). More details on how to use these datasets can be found at: França et al., (2021) Historical Data in the CoastNet Geoportal: Documenting Fish Assemblages in Portuguese Estuaries. Front. Mar. Sci. *8*, 685294. https://doi.org/10.3389/fmars.2021.685294.

**Acknowledgments:** Historical data used in the present work were collected during the 3 following research projects: (1) ERIC-Effects of river flow changes on the fish communities of the Douro, Tagus and Guadiana estuaries and adjoining coastal areas: Ecological and socio-economical predictions (PDCTM/C/MAR/15263/1999); (2) NURSERIES—Importance of estuarine and coastal nursery areas for fish stocks of commercially valuable species in the Portuguese coast (FEDER 22-05-01-FDR-00037-Programa MARE, 2005–2006), supported by FEDER; (3) FISHEST–Spatial and temporal variability of fish assemblages in different estuarine habitats: dependence, adaptive plasticity, and resilience (PTDC/MAR/64982/2006), supported by Fundação para a Ciência e a Tecnologia. The author would like to acknowledge all the researchers from these projects' teams involved in the sampling, laboratory and data analysis; and researchers from the CoastNet team who prepared the data to be integrated in the CoastNet Geoportal and contributed to the research infrastructure implementation and functioning, namely the historical data section.

**Conflicts of Interest:** The author declares no conflict of interest. The funders had no role in the design of the study; in the collection, analyses, or interpretation of data; in the writing of the manuscript; or in the decision to publish the results.

## Appendix A

**Table A1.** List of the fish species caught in the three projects and respective ecological guild, according to Franco et al. (2008) [46] (C—Catadromous Species; ES—Estuarine Species; MS—Marine Stragglers; MM—Marine Migrants; F—Freshwater Species).

| Species | Ecological Guild |
| --- | --- |
| *Anguilla anguilla* | C |
| *Aphia minuta* | ES |
| *Argyrosomus regius* | MS |
| *Arnoglossus imperialis* | MS |
| *Arnoglossus laterna* | MS |
| *Atherina boyeri* | ES |
| *Atherina presbyter* | MM |
| *Belone belone* | MS |
| *Boops boops* | MS |
| *Bothus podas* | MS |
| *Buglossidium luteum* | MS |
| *Callionymus lyra* | MS |
| *Callionymus reticulatus* | MS |
| *Chelidonichthys lucernus* | MS |
| *Chelidonichthys obscurus* | MS |
| *Chelon auratus* | MS |
| *Chelon labrosus* | C |
| *Chelon ramada* | C |
| *Conger conger* | MS |
| *Dicentrarchus labrax* | MM |
| *Dicentrarchus punctatus* | MM |
| *Dicologlossa cuneata* | MM |
| *Dicologlossa hexophthalma* | MS |
| *Diplodus annularis* | MM |

**Table A1.** *Cont.*

| Species | Ecological Guild |
|---|---|
| *Diplodus bellottii* | MM |
| *Diplodus sargus* | MM |
| *Diplodus vulgaris* | MM |
| *Echiichthys vipera* | MS |
| *Engraulis encrasicolus* | MS |
| *Gobius niger* | ES |
| *Gobius paganellus* | ES |
| *Halobatrachus didactylus* | ES |
| *Hippocampus hippocampus* | ES |
| *Luciobarbus bocagei* | F |
| *Merluccius merluccius* | MS |
| *Mullus surmuletus* | MS |
| *Pagellus acarne* | MS |
| *Pagellus bagaravea* | MS |
| *Pagrus auriga* | MS |
| *Pagrus pagrus* | MS |
| *Pegusa lascaris* | MS |
| *Platichthys flesus* | MM |
| *Pollachius pollachius* | MS |
| *Pomastoschistus microps* | ES |
| *Pomastoschistus minutus* | ES |
| *Raja clavata* | MS |
| *Raja microocellata* | MS |
| *Raja montagui* | MS |
| *Raja undulata* | MS |
| *Sardina pilchardus* | MS |
| *Scomber scombrus* | MS |
| *Scophthalmus maximus* | MM |
| *Scophthalmus rhombus* | MM |
| *Scorpaena notata* | MS |
| *Solea senegalensis* | MM |
| *Solea solea* | MM |
| *Sparus aurata* | MS |
| *Spondyliosoma cantharus* | MS |
| *Sprattus sprattus* | MS |
| *Symphodus bailloni* | MS |
| *Syngnathus abaster* | ES |
| *Syngnathus acus* | ES |
| *Syngnathus typhle* | ES |
| *Trachurus trachurus* | MS |
| *Trisopterus luscus* | MS |
| *Zeus faber* | MS |

## Appendix B

**Table A2.** Specifications of the models used in the BIOMOD2 package (adapted from Thuiller et al. [50]).

| Model | Specifications |
|---|---|
| GLM | Run a stepwise GLM using linear ("simple") terms. The statistical criteria used for selection of models of increasing fit were the Akaike Information Criterion (AIC). To select the most parsimonious model, BIOMOD uses an automatic stepwise model selection. |

**Table A2.** *Cont.*

| Model | Specifications |
|---|---|
| GAM | Run a generalized additive model with a spline function with a degree of smoothing of 4 (similar to a polynomial of degree 3).<br>BIOMOD uses a cubic spline smoother, which is a collection of polynomials of degree less than or equal to 3, defined on subintervals delimited by knots. A separate polynomial, fitted for each neighborhood, enables the fitted curve to join all the points, producing a smooth linear curve. BIOMOD uses an automated stepwise process to select the most significant variables for each species. Y = s(X1, 4) + s(X2, 4) + s(X3, 4). |
| RF | Run a random forest model. Implements Breiman's random forest algorithm (based on Breiman and Cutler's original Fortran code) for classification and regression. It is implemented into the "random-Forest" library (Liaw and Wiener). BIOMOD uses 500 trees and extracts the importance of each selected variable. |
| CTA | Run a classification tree analysis (CTA). The optimal length of the tree is estimated using cross-validation (default = 50). BIOMOD uses a procedure running k-fold cross-validations to select the best tree, which is a trade-off between the number of leaves of the tree and the explained deviance. BIOMOD uses the rpart library to run the classification tree analysis. |

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
