# Peer review of "Changes in the Functional Role of the Tejo Estuary (Portugal, Europe) According to Fish Ecological Guilds"

_fishes, doi:10.3390/fishes8110545_

Round 1
Reviewer 1 Report
Comments and Suggestions for Authors
The article entitled “Changes in the functional role of the Tejo estuary according to 2 fish ecological guilds: what to expect in the near future?” uses the guild approach to functionally describe the fish structure of the Tejo Estuary and model the possible outcomes of future environment changes.
Although I believe the article deals with an important topic in a proper, the paper needs some modifications to be read for publication.
In the introduction section, the main problem is that there are a lot of information that, sometimes, take the text fluidity due to the way the text is formatted. For instance, the third paragraph is huge, and talks about many topics, from biotic composition of estuaries to functional role of fish species, as well as the use of guilds to comprehend patterns, and how climate changes may influence the current dynamics. I think that this section needs some edit to make readers understand the main goal of the paper.
Another issue that caught my attention is that the author states, in the method section, that marine migrant and estuarine resident fishes are the most represented ecological guilds (line 196). However, during the results section, the marine stragglers (MS) are the most representative in terms of species richness, the estuarine residents are the most abundant guild, and the freshwater (F) guild had the highest values of biomass. I would like to understand how the most represented ecological guilds were chosen, and why the MS and F guilds were not included in the models described in the paper.
Since recent works have shown that MS species play a significant role in estuarine areas, due to their ability to enhance niche differentiation (see da Silva and Fabré, 2019; Estuaries and Coasts), I believe that modelling how they would suffer from climate change would be amazing for the paper, since the guild has so many species occurring in the estuary.
Reviewer 2 Report
Comments and Suggestions for Authors
I like this paper. In my opinion, the doc is correctly written, the analyses are adequate for the aim the author pretends to achieve, and the study is globally consistent. I believe that I consider this a good study.
Only a few minor changes:
Line 10: yes, this is a relevant issue. I suggest this paper regarding this matter:
Flecker, A. S., McIntyre, P. B., Moore, J. W., Anderson, J. T., Taylor, B. W., & Hall Jr, R. O. (2010). Migratory fishes as material and process subsidies in riverine ecosystems. In American Fisheries Society Symposium (Vol. 73, No. 2, pp. 559-592).
Lines 144-160: please, can you include the number of samplings? I am a little bit confused about it. There are a lot of hauls, projects and samplings… perhaps the global data about the number of considered samplings is useful for the reader. Thanks.
Line 214: change “4” to “four”.
Line 248: change “3” to “three”.
Line 359: delete the comma.
Round 2
Reviewer 1 Report
Comments and Suggestions for Authors
The author has improved its work and I am very satisfied with the current version.
Author Response
Thank you very much for taking the time to review this manuscript. The reviewer was very satisfied with the current version.
I took the opportunity to rephrase several parts of the manuscript, as suggested.
Thank you,
Susana França